# Identification of Novel Mutations by Targeted NGS Panel in Patients with Hyperferritinemia

**DOI:** 10.3390/genes12111778

**Published:** 2021-11-09

**Authors:** Giulia Ravasi, Sara Pelucchi, Francesca Bertola, Martina Maria Capelletti, Raffaella Mariani, Alberto Piperno

**Affiliations:** 1Department of Medicine and Surgery, University of Milano-Bicocca, 20900 Monza, Italy; giulia.ravasi@unimib.it (G.R.); sara.pelucchi@unimib.it (S.P.); m.capelletti@campus.unimib.it (M.M.C.); 2Medical Genetics, S. Gerardo Hospital, ASST-Monza, 20900 Monza, Italy; f.bertola@asst-monza.it; 3Disorders of Iron Metabolism, Centre for Rare Diseases, San Gerardo Hospital, ASST-Monza, 20900 Monza, Italy; r.mariani@asst-monza.it

**Keywords:** ferroportin, transferrin receptor 2, hepcidin, hemojuvelin, next generation sequencing, hemochromatosis, ferritin, iron overload

## Abstract

Background. Several inherited diseases cause hyperferritinemia with or without iron overload. Differential diagnosis is complex and requires an extensive work-up. Currently, a clinical-guided approach to genetic tests is performed based on gene-by-gene sequencing. Although reasonable, this approach is expensive and time-consuming and Next Generation Sequencing (NGS) technology may provide cheaper and quicker large-scale DNA sequencing. Methods. We analysed 36 patients with non-*HFE*-related hyperferritinemia. Liver iron concentration was measured in 33 by magnetic resonance. A panel of 25 iron related genes was designed using SureDesign software. Custom libraries were generated and then sequenced using Ion Torrent PGM. Results. We identified six novel mutations in *SLC40A1*, three novel and one known mutation in *TFR2*, one known mutation and a de-novo deletion in *HJV*, and a novel mutation in *HAMP* in ten patients. In silico analyses supported the pathogenic role of the mutations. Conclusions. Our results support the use of an NGS-based panel in selected patients with hyperferritinemia in a tertiary center for iron metabolism disorders. However, 26 out of 36 patients did not show genetic variants that can individually explain hyperferritinemia and/or iron overload suggesting the existence of other genetic defects or gene-gene and gene-environment interactions needing further studies.

## 1. Introduction

There are several inherited diseases causing hyperferritinemia with or without iron overload, and with high or normal-low transferrin saturation (TSAT) [1]. Hyperferritinemia with iron overload includes hemochromatosis (HH) due to mutation in *HFE, HJV*, *HAMP*, *TFR2* and *SLC40A1* gain-of-function mutations, hereditary iron loading anemias, and hypo-transferrinemia that are characterized by high TSAT [2]. Aceruloplasminemia and Ferroportin disease due to loss-of-function mutation of *SLC40A1*, commonly show iron overload and low/normal TSAT. Hereditary hyperferritinemia-cataract syndrome (HHCS) and benign hyperferritinemia, respectively, due to mutations in the iron responsive element (IRE) of the L-ferritin gene (*FTL*) and in the first exon of *FTL*, have normal TSAT and no iron overload. In addition, there are many acquired forms of hyperferritinemia and secondary causes of iron overload that can further complicate the big picture. Thus, a differential diagnosis of hyperferritinemia requires strategies including family and personal medical history, laboratory tests, non-invasive quantification of liver iron concentration, and eventually genetic tests [1,2,3]. According to TSAT levels, patients should be addressed to different diagnostic step-by-step pathways to define the genetic or acquired cause of hyperferritinemia and iron overload and choose the best therapeutic approach. Type 1 HH, most commonly due to p.Cys282Tyr homozygous mutations in *HFE*, is the most frequent inherited iron overload disease, but is characterized by incomplete penetrance and variable expression. In few cases, rare *HFE* mutations in the compound heterozygous state with p.Cys282Tyr, and single mutations in non-*HFE* genes associated to the p.Cys282Tyr homozygous genotype (digenic HH) may lead to full penetrant and severely expressed phenotypes [4,5]. Non-*HFE* HH includes type 2A, 2B, and 3 that are rare autosomal recessive disorders caused by different and often private mutations in *HJV*, *HAMP*, and *TFR2*, respectively. Type 2A and 2B HH, usually referred as juvenile HH, are characterized by early and severe iron-related complications, while type 3 HH is generally considered a form with intermediate severity. Mutations in *SLC40A1*, coding for Ferroportin, can lead to different autosomal dominant phenotypes: gain-of-function mutations (*SLC40A1*^gf^) are responsible for type 4 HH (formerly HH Type 4B) whose manifestations resemble those of HFE-HH, while the more frequent loss-of-function-mutations (*SLC40A1^lf^*) lead to Ferroportin disease (formerly HH Type 4A) often characterized by hyperferritinemia with normal TSAT and prevalent sinusoidal iron accumulation [1,6,7]. According to this classification, the different types of HH are all characterized by increased intestinal iron absorption and release of iron from macrophages, high TSAT, increased serum ferritin, and prevalent iron accumulation in parenchymal cells. However, they may differ for the age of presentation, rate and amount of iron accumulation and clinical manifestations. In addition, recent evidences indicate that phenotypes may overlap among different types of HH [1]. Thus, definite diagnosis may require sequencing of each potentially causative gene (*HJV*, *HAMP* and *TFR2*, but also *SLC40A1* and *HFE*), which is expensive, and time-consuming [8].

The other inherited disorders have also distinctive phenotypes: inherited iron-loading anemias are characterized by chronic anemias [2], hereditary hypo-transferrinemia by very low serum transferrin [1], aceruloplasminemia by the lack of measurable serum ceruloplasmin [9], and HHCS by early-onset cataract in the proband and his/her relatives, although there is a high variability in the time of onset of the cataract and its severity [9,10]. Conversely, Ferroportin disease and benign hyperferritinemia are not easily distinguishable each other and with the frequent forms of acquired hyperferritinemia associated with metabolic abnormalities, overweight and liver steatosis as they are characterized by hyperferritinemia with normal TSAT, variable liver iron concentration (LIC) from normal to mild-moderate iron overload (often with prevalent sinusoidal or mixed distribution) [6,11,12,13]. Thus, even in these cases multiple tests might be required to define the right diagnosis.

Next Generation Sequencing (NGS) technology overcomes the restriction of single gene analysis and could provide cheaper and more rapid large-scale DNA sequencing [14]. Here we described the results of an NGS-based custom panel to simultaneously analyze 25 genes involved in iron metabolism in patients with hyperferritinemia referring to a tertiary center for disorders of iron metabolism.

## 2. Materials and Methods

### 2.1. Patients

We selected 36 unrelated patients with hyperferritinemia who presented to the Centre for Disorder of Iron Metabolism at the ASST-Monza, S.Gerardo Hospital because of the suspicion of non-HFE HH, Ferroportin disease or inherited hyperferritinemia. According to this aim we previously excluded patients if they were: i. homozygous for the p.Cys282Tyr or compound heterozygous for the p.Cys282Tyr and p.His63Asp in *HFE*; ii. affected by acquired causes of hyperferritinemia: e.g., dysmetabolic alterations, chronic liver diseases of any etiology (alcohol-, viral-, autoimmune-related); iii. affected by iron loading anemias (transfusion and not-transfusion dependent thalassemia, congenital dyseritropoietic or sideroblastic anemias, myelodisplastic syndrome); iv. transfused in the past; v. affected by Gaucher’s disease.

The selected patients underwent to clinical evaluation, follow-up and therapies as needed. They were addressed to targeted NGS-based panel test aiming to identify causal mutations and gene variants that might contribute to their hyperferritinemia/iron overload phenotype. All patients gave their written informed consent for genetic testing according to the Institutional Review Board of ASST-Monza, S. Gerardo Hospital—Monza.

Demographic data, biochemical and magnetic resonance (MRI) values were those at diagnosis. Biochemical tests were performed by commercial kits. Liver iron quantification by MRI (LIC^MRI^) was performed according to Galimberti et al. [15] in 33 patients. Three patients refused or were not able to perform MRI evaluation because of claustrophobia.

### 2.2. Panel Design

An NGS-based panel was designed to analyze genotypes of the 36 patients. The panel included 25 genes currently associated with iron metabolism and genetic iron overload. It was designed and optimized using SureDesign software (Agilent Technologies, Santa Clara, CA, USA). It consists of 12,224 amplicons with an average of 200 bp in length, covering a cumulative target sequence of 185,730 kbp including coding regions, exon/intron junctions, promoter and target regions of SNPs of interest. The complete list of the genes is reported in Appendix A. 

### 2.3. Library Preparation and Sequencing

Genomic DNA was isolated from fresh EDTA blood samples and DNA concentration was measured using Qubit dsDNA assay kit with Qubit fluorometer 2.0 (Thermo Fisher Scientific, Waltham, MA, USA). Custom libraries were generated starting from 225 ng DNA using a custom design HaloPlex Target Enrichment kit (Agilent Technologies, Santa Clara, CA, USA) according to manufactures protocol. Libraries quality and quantity were determined using the Agilent High Sensitivity DNA kit on the Agilent 2100 bioanalyzer (Agilent Technologies, Santa Clara, CA, USA). Libraries were then sequenced using Ion PGM Hi-Q OT2 kit and Ion PGM Hi-Q Sequencing Kit on Ion PGM 318 chip V2 on an Ion Torrent PGM (Thermo Fisher Scientific, Waltham, MA, USA).

### 2.4. Analysis

Post run QC/QA filtering was performed using Torrent Suit (v3.6; Thermo Fisher Scientific, Waltham, MA, USA), and sequences aligned to the human genome version 19 (HG19). Variants were called using the Torrent Variant Caller (version 5.2).

Genetic variants were considered novel when not annotated in HGMD professional. Novel variants were named according to the Human Genome Variation Society (HGVS) recommendations (http://www.hgvs.org/mutnomen/recs.html, accessed on 20 October 2021). Bioinformatics tools were used to predict their deleterious effect. Multiple sequence alignment of missense variants and their putative effect on protein structure and function, were analysed using the softwares Polyphen-2 (http://genetics.bwh.harvard.edu/pph2/index.shtml, accessed on 20 October 2021), SIFT (http://sift.bii.a-star.edu.sg/www/SIFT_seq_submit2.html, accessed on 20 October 2021), MutPred (http://mutpred.mutdb.org/, accessed on 20 October 2021) and Mutation Taster (http://www.mutationtaster.org/, accessed on 20 October 2021). Splicing variants were studied using Human Splicing Finder, vs. 3.0 (http://www.umd.be/HSF3/HSF.html, accessed on 20 October 2021). In Table 1 we calculated *p* values using Chi-squared test by comparing the frequencies of wild-type and mutated alleles in the patients (taken as whole or divided according to TSAT) with those registered in gnomAD database.

### 2.5. Multiplex Ligation-Dependent Probe Amplification (MLPA) Assay

MLPA was performed using the Salsa MLPA probemix P347-A3 according to the standard MLPA protocol (MRC-Holland, Amsterdam, The Netherlands). Coffalyser software (coffalyser.net, accessed on 20 October 2021) was used for analysis of peak values obtained from capillary electrophoresis on ABI 3500 instrument (Applied Biosystems, Waltham, MA, USA). Mean cut-off for normalized peak height ratio of patient to the control sample was less than 0.7 in case of deletions and more than 1.30 in case of duplications.

### 2.6. Generation of SLC40A1 Mutant Construct

5′-UTR region of Ferroportin gene was cloned in pGL4.23 reporter vector (Promega Corp., Madison, WI, USA) upstream of the firefly luciferase gene by using NcoI Restriction Enzyme and Ligation-Free Cloning System (Applied Biological Materials Inc. Richmond, BC, Canada) with sequence specific primers (forward primer: 5′-GGTACTGTTGGTAAAGCCACGGGACGCCCGGGCGGC-3′ and reverse primer: 5′-TGTTTTTGGCATCTTCCATGGACACTAGGCGACCCCGC-3′) and human genomic DNA as template, according to manufacturer’s protocol. Plasmid DNA was isolated by innuPREP Plasmid Mini Kit Plus (Analytik Jena AG, Jena, Germany) for transfection. To generate mutant construct, site direct mutagenesis was performed using the Q5^®^ Site-Directed Mutagenesis Kit (New England BioLabs Inc., Ipswich, MA, USA) according to the manufacturer’s protocol and using the specific oligonucleotides (forward primer: 5′- GTGTTAGCTAcGTTTGGAAAG -3′ and reverse primer: 5′- TGTAGCTGAAGTTGGAAAG -3′). All vectors were verified by direct sequencing.

### 2.7. Cell Culture and Dual-Luciferase Reporter Assay

The human cell line HeLa was grown in Dulbecco Modified Eagle Medium (DMEM) supplemented with 10% heat-inactivated fetal bovin serum (FBS), glutamine and combined antibiotics, at 37 °C and 5% CO_2_. A pGL4.23 basic reporter vector (Promega Corp., Madison, WI, USA) harboring a 427bp fragment of human Ferroportin 5′-UTR was used to analyze luciferase activity. HeLa cells were seeded in 48-well plates at 40% of confluency. Cells were transiently co-trasfected by Fugene HD transfection reagent (Promega Corp., Madison, WI, USA) with Ferroportin 5′-UTR wild type or mutated construct (250 ng) and pRL-TK renilla lulciferase vector (15 ng) (Promega Corp., Madison, WI, USA) to control transfection efficiency. The pGL4.23 basic plasmid was also co-transfected with pRL-TK renilla as a negative control. After 48 h cells were lysed and firefly and renilla luciferase activities were measured by a Glomax Multi JR luminometer according to manufacturer’s protocols (Promega Corp., Madison, WI, USA). Relative luciferase activity was calculated as the ratio between firefly (reporter) and Renilla luciferase activity. Each construct was tested in triplicate, and the transfection experiments were performed three times independently. Luciferase activities were compared by the Mann-Whitney test, using Prism 3.2 software (GraphPad Software, San Diego, CA, USA).

## 3. Results

Of the 36 subjects with hyperferritinemia, 25 had TSAT levels < 45% and 11 ≥ 45%, confirmed by at least two tests. Table 1 shows the main data of the patients divided according to TSAT levels. They do not differ for gender, body mass index, age at diagnosis, alcohol intake, hemoglobin, and liver function tests. Iron parameters and serum ALT were significantly higher in patients with TSAT ≥ 45%. Appendix A shows the 51 exonic non-synonymous variants and their allele frequencies according to TSAT compared with gnomAD ones, while in Table 2 we report only the variants (novel or already known) predicted as probably damaging by in silico tools. Appendix A shows the distribution of these variants (in the heterozygous or homozygous state) in each patients. There were 12 patients with HFE variants, one p.Cys282Tyr heterozygote, 5 p.His63Asp heterozygotes and 4 homozygotes, and two p.Ser65Cys heterozygotes without differences between high and normal TSAT groups. The frequency of p.His63Asp allele significantly differed between patients and an Italian healthy control group (0.119; *p* = 0.02) [16] and gnomAD (*p* = 0.003). In particular, p.His63Asp homozygotes were significantly more represented both in the whole group (0.111; *p* < 0.005) and in the TSAT <45% group (0.12; *p* < 0.005) compared to Italian healthy controls [16] (0.014). Although higher than in the control group, the frequency of p.His63Asp homozygotes in TSAT ≥ 45% group (0.091) did not reach the statistical significance. Of the four patients homozygotes for the p.His63Asp variant, two showed also *SLC40A1* mutations (see below). NGS identified causal mutations in non-*HFE* HH genes in ten unrelated patients (25.6%): six novel mutations in *SLC40A1*, four mutations (three novel and one already known [17]) in *TFR2* in 2 patients in the compound heterozygous state, one novel mutation in *HJV* in compound heterozygosity with a de novo *HJV* deletion, and one novel mutation in *HAMP*.

Iron status and mutations of the 10 patients are briefly described below and shown in Table 3 and Table 4, respectively. All the six patients with *SLC40A1* mutations had slightly or moderately increased iron overload, normal hepatic function tests, liver ultrasound and elastography, excluding patient 3 and 5 who showed mild steatosis; patient 3 showed also metabolic abnormalities at diagnosis (high triglycerides and low HDL); patient 5 underwent to liver biopsy that showed normal liver structure, moderate iron overload as assessed by Deugnier’s score [18] with mixed (hepatic and reticuloendothelial cells) hemosiderin iron distribution. Patients 4 and 6 were also homozygotes for the p.His63Asp variant in HFE; patient 6 was also heterozygous for p.Pro477Leu and patient 2 was compound heterozygote for p.Thr511Ile and p.Pro876Ser variants in *CP* gene although ceruloplasmin concentrations were in the normal range in both (21 and 19 mg/dL, respectively; n.v. 15–30 mg/dL). Patients 1, 2, 3, and 5 were phlebotomized and iron depleted after removal of 3.4, 2.0, 4.0, and 3.2 g of iron, respectively. Family screening revealed other first-degree relatives with hyperferritinemia that carried probands’ mutations in agreement with the dominant phenotype.

Patients 7 and 8 carried *TFR2* mutations in the compound heterozygous state. As expected, they showed very high TSAT. Patient 7 did not show liver iron overload yet, likely due to his young age. Patient 8 showed increased serum ALT and marked iron overload at MRI; liver biopsy showed marked hemosiderin deposits with prevalent hepatocellular distribution, mild steatosis, and mild fibrosis (grade 1 according to Ishak et al. [19]). No other affected relative was found.

Two patients showed phenotypes supporting a juvenile (patient 9) and juvenile-like hemochromatosis (patient 10). Patient 9 presented with a severe heart failure and paroxysmal atrial fibrillation requiring urgent hospitalization in intensive care unit in another hospital in Milan. Echocardiography showed severe biventricular dysfunction (ejection fraction 28%), pleural effusion and ascites. The diagnostic work-up showed mild normocytic anemia (Hb 12.3 g/dL), renal insufficiency and alteration of liver function tests, marked hyperferritinemia (3510 ng/mL) and TSAT (92%). Liver and cardiac magnetic resonance showed marked liver (T2* 1.02 msec) and heart (T2* 5.14 msec) iron overload. After consulting, combined iron chelation therapy (subcutaneous infusion of deferoxamine 40 mg/kg/day, and deferiprone 75 mg/kg/day) was begun, and a sample sent to our laboratory for genetic testing that confirmed the diagnosis of HJV-related hemochromatosis. In the following weeks the patient showed significant clinical and cardiac improvement and reduction of iron overload. An implantable cardioverter defibrillator (ICD) for prevention of fatal arrythmias was placed. The patient has now returned to his normal activities and is followed up on an outpatient basis. Patient 10 was sent to our attention because of markedly high serum ferritin and TSAT firstly discovered during investigations for Cytomegalovirus infection, and repeatedly confirmed thereafter. She showed marked liver but not heart iron overload at MRI, normal glucose, hepatic and gonadal function tests.

Details of the mutations in the 10 patients and in silico prediction are reported in Table 4. To notice, p.Arg179Thr in *SLC40A1* and p.Tyr547Phe in *TFR2* are reported in ExAC (rs765023388 and rs573769443, respectively) in a single subject in the heterozygous state. p.Arg455Gln mutation in *TFR2* was already described in an affected patient [17]. All the other mutations are novel. MLPA testing showed the presence of de novo *HJV* deletion in compound heterozygosity with p.Cys89Arg in patient 9. In patient 10 we identified a single novel mutation in *HAMP* that she shared with her father and brother showing normal iron indices suggesting that another mutation in another gene inherited from her mother exists. Whole exome sequencing of the proband and her parents is under evaluation.

Because of the peculiar location of the c.-205A>C mutation in the 5′UTR of the *SLC40A1* iron responsive element (IRE), we performed a functional analysis. According to Mutation Taster, this variant was predicted as disease causing. To evaluate whether and how this mutation could affect ferroportin expression, we transfected HeLa cells with WT or mutated *SLC40A1* 5′UTR luciferase constructs. Results of luciferase experiments are reported in Figure 1. Transfection with mutated construct led to a significant upregulation of luciferase activity compared to WT one (*p* < 0.0001).

In the other 26 patients (19 with TSAT < 45% and 7 with TSAT ≥ 45%) we identified several probably damaging variants mostly in the heterozygous state as reported in Table 2 and Appendix A. Liver iron quantification by MRI, performed in 24 showed mild to moderate iron overload (median 52.4 μmol/g; 1st and 3rd quartiles 37.4–75.8). The p.His63Asp variant was the commonest with an allele frequency of 0.173 not significantly different compared to an Italian control group [16]. There are two homozygotes (0.077, at the border of significance *p* = 0.056). Among the other variants, only p.Leu96Pro in *BMP6*, found in two patients in the heterozygous state, was significantly more represented in patients with TSAT < 45% than in gnomAD (Table 2). Overall, 7 patients did not carry any of the reported variants while the other have variable combinations ranging from a single (5 patients) to 4 variants (1 patients). Last, two patients carried the β-Thalassemia trait with a frequency (0.077) slightly higher than in general population in Italy (Appendix A).

## 4. Discussion

In this study, we illustrate the results of an NGS-based targeted gene panel to identify genetic cause in 36 cases of patients with non-HFE related hyperferritinemia. Current recommendations suggest a stepwise approach (single-gene testing) in which sequencing should be made initially according to clinical features (i.e., prioritizing *HJV* and *HAMP* in patients with early-onset and severe iron overload) [8]. While this is reasonable, it is time consuming and does not take appropriately into account the emerging phenotype overlaps between different forms of HH [20,21,22,23], the possible occurrence of digenic inheritance or the presence of rare and severe *HFE* mutations, and the variable phenotype of ferroportin mutation [1]. Previous studies demonstrated the feasibility, sensitivity and cost-effectiveness of NGS-targeted gene panels and exome sequencing in patients suspected for non-HFE hemochromatosis and Ferroportin disease [24,25,26,27]. In addition, NGS-based testing reduced turnaround time of analysis and clinical reporting time. Accordingly, using an NGS-targeted gene panel we were able to detect causal mutations in non-HFE genes in ten unrelated patients (27.8%) by a single run and to draw a diagnostic report in few days including those required to confirm the presence of mutations by Sanger sequencing. 

Multiple mutations in *SLC40A1* gene have been described so far [6,7,27]: loss-of-function variants are characterized by diminished cell surface expression, lower iron export capacity, and hyperferritinemia with normal TSAT (Ferroportin disease), while gain-of-function variants are characterized by ferroportin resistance to hepcidin activity resulting in continued iron export and a hemochromatosis-like phenotype (Hemochromatosis type 4) [6,27,28]. All the mutations that we identified, but c.-205A>C mutation in the IRE region, are classified as deleterious by in silico studies (Table 4), and patients’ phenotypes suggest that they behave as loss-of-function mutations as indicated by the high serum ferritin and normal TSAT levels (Table 3). Our findings confirm that Ferroportin disease is likely the most common form of non-HFE inherited iron overload disorder and that is generally characterized by mild or moderate iron overload and benign outcome [6].

The c.-205A>C mutation located into the stem of the IRE in the 5′-UTR of *SLC40A1* mRNA (7 bp after the CAGUGU), is the first report of a mutation in the *SLC40A1* IRE region. Liu et al. [29] previously described a 43-year-old Japanese woman with a mutation located just after the IRE region (c.-181A>G) in the heterozygous state. The Japanese patient showed markedly high ferritin and TSAT levels, and severe liver iron overload complicated by hepatic alterations and glucose intolerance. Whether this mutation affected the iron regulatory protein (IRP) interaction with the *SLC40A1* IRE causing post-transcriptional ferroportin up-regulation [30,31] and a hemochromatosis-like phenotype is not clear, as functional study was not performed and juvenile hemochromatosis genes were not sequenced. Conversely, in vitro experiments showed that the c.-205A>C mutation led to a significant up-regulation of luciferase activity compared to WT one (*p* < 0.0001). This suggests that, as observed in HHCS for L-ferritin [12,31], the mutation could lead to constitutive ferroportin expression, larger availability of ferroportin molecules on the cellular membrane and greater efflux of iron from cells to plasma. However, patient clinical manifestations characterized by normal TSAT, mild iron overload with mixed hemosiderin accumulation in both hepatocytes and sinusoidal cells do not support what observed in vitro and raise doubts on the effect of the mutations in the clinical setting.

At the present, 54 mutations in *TFR2* are listed in The Human Gene Mutation Database (HGMD^®^) Professional 2014.2. (Institute of Medical Genetics in Cardiff. Available from: http://www.hgmd.cf.ac.uk/ac/index.php, accessed on 20 October 2021) with variable phenotype expression. NGS allowed identifying 3 novel and 1 already known mutations presenting in the compound heterozygous state in 2 male patients. The causative role of these mutations is supported by the severe iron overload phenotype of patient 10, the high TSAT and HII of patient 9, and by in silico studies that classified the mutations as likely deleterious (Table 4).

A short comment deserves the findings on *HFE* p.His63Asp homozygosity. Although generally considered a very low penetrant genotype with controversial effect on iron homeostasis [8,32] it was found to be more represented in patients compared to healthy controls. In two patients it coexists with *SLC40A1* mutations, but it was unable to modify the Ferroportin disease phenotype characterized by hyperferritinemia with normal TSAT, and in the other two patients it was associated with hyperferritinemia alone. These findings support the low penetrance of the p.His63Asp genotype, but suggest that it might contribute to genetic susceptibility to hyperferritinemia and mild-moderate iron overload.

In the clinical practice, some individuals with hyperferritinemia and mild-moderate iron overload and, less frequently, with severe iron overload remain without a specific genetic diagnosis. Accordingly, twenty-six patients in this series (72%) mostly with mild-moderate iron overload did not show genetic variants that can individually explain hyperferritinemia and iron overload. Some showed one or more variants in the heterozygous state in genes involved in iron metabolism (Appendix A) suggesting a polygenic background of susceptibility, but these findings are far from being used in diagnostics. Furthermore, the absence of genetic variants in some of them might support the existence of other genetic and/or acquired disorders mimicking mild forms of hemochromatosis in patients with hyperferritinemia with TSAT ≥ 45% or FTL-related diseases and Ferroportin diseases in patients with hyperferritinemia with normal TSAT.

## 5. Conclusions

In conclusion, our findings support the use of NGS in identifying genetic defects in patients suspected for non-*HFE* hemochromatosis, and in accurately selected patients with hyperferritinemia according to literature indications [1,8,11,12]. However, in many subjects we were not able to identify genetic variants able to explain hyperferritinemia and iron overload. Other genetic defects still to be discovered might be implicated as suggested by the recent description of cases with unexplained hyperferritinemia possibly inherited as a recessive trait [21]. By the other hand, it is possible that in some cases hyperferritinemia and/or iron overload are complex phenotype and that analyzing a single locus is not enough to understand their pathophysiology. Thus, it is possible that complex gene-gene and gene-environment interactions exist, as hypothesized in the more common complex diseases such as autism, hypertension and diabetes [33] that need to be further clarified.

## Figures and Tables

**Figure 1 genes-12-01778-f001:**
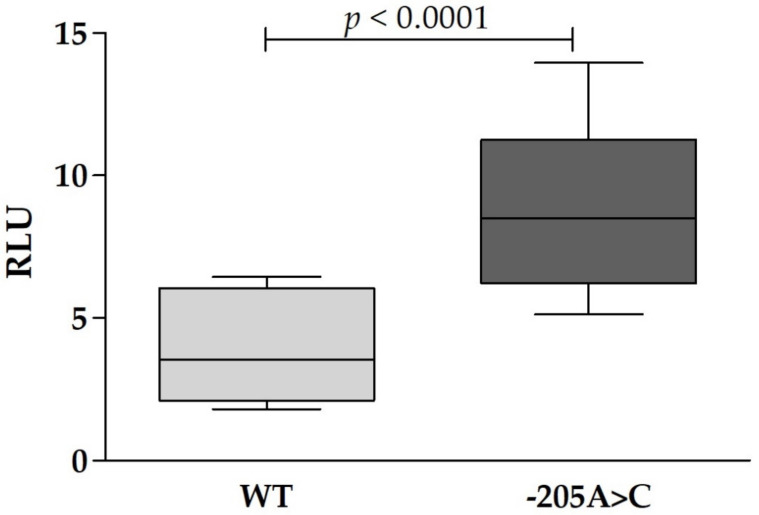
Luciferase assay in HeLa cell line transfected with WT or mutated ferroportin 5′UTR luciferase constructs. Boxes denote lower quartile, mean and upper quartile, and whiskers show maximum and minimum ranges. (RLU = Relative Luciferase Unit).

**Table 1 genes-12-01778-t001:** List of 30 variants (including novel mutations) and their allelic frequencies found in the whole 36 patients, and in subgroups according to transferrin saturation (TSAT) compared to gnomAD frequencies.

Gene	Amino Acid Change	dbSNP	Allelic Frequency (All)	Allelic Frequency (TSAT < 45%)	Allelic Frequency (TSAT ≥ 45%)	Allelic Frequency (gnomAD)	*p* (All vs. gnomAD)
*ACO1*	V498M	rs375879049	0.014	0.000	0.045	0.000	ns
*BMP6*	L96P	rs200573175	0.028	0.040	0.000	0.006	0.01
*CDAN1*	R891C	rs8023524	0.111	0.120	0.091	0.186	ns
*CP*	P477L	rs35331711	0.014	0.020	0.000	0.004	ns
*CP*	P876S	rs767188986	0.014	0.000	0.000	0.000	ns
*CP*	T551I	rs61733458	0.042	0.060	0.000	0.031	ns
*ERFE*	G196R	rs1247409936	0.014	0.020	0.000	0.000	ns
*HAMP*	K83delinsLIYSCCPR	rs1159254691	0.014	0.020	0.000	0.000	ns
*HAMP*	R59P	novel	0.014	0.000	0.045	//	//
*HJV*	C89R	novel	0.014	0.000	0.045	//	//
*HJV*	Deletion	novel	0.014	0.000	0.045	//	//
*HIF1A*	P582S	rs11549465	0.097	0.080	0.136	0.105	ns
*PCSK7*	T182M	rs150784623	0.014	0.000	0.045	0.000	ns
*SERPINA1*	E366K	rs28929474	0.014	0.000	0.045	0.000	ns
*SERPINA1*	D280V	rs121912714	0.014	0.020	0.000	0.001	ns
*SLC40A1*	V160A	novel	0.014	0.020	0.000	//	//
*SLC40A1*	R179T	rs765023388	0.014	0.020	0.000	0.00001	<0.001
*SLC40A1*	A350D	novel	0.014	0.020	0.000	//	//
*SLC40A1*	G494D	novel	0.014	0.020	0.000	//	//
*SLC40A1*	c.-205A>C	novel	0.014	0.020	0.000	//	//
*SLC40A1*	V531A	novel	0.014	0.020	0.000	//	//
*SMAD4*	N188D	novel	0.014	0.020	0.000	//	//
*SMAD7*	G137A	rs757109163	0.014	0.000	0.045	0.000	ns
*SMAD7*	G39R	rs144204026	0.028	0.040	0.000	0.011	ns
*TF*	G544E	rs121918677	0.014	0.020	0.045	0.000	ns
*TF*	G150S	rs1799899	0.083	0.080	0.091	0.060	ns
*TFR2*	L224R	novel	0.014	0.000	0.045	//	//
*TFR2*	D514Mfs12	novel	0.014	0.000	0.045	//	//
*TFR2*	R455N	rs41303501	0.014	0.000	0.045	0.00342	ns
*TFR2*	Y547F	rs573769443	0.014	0.000	0.045	0.00009	<0.001

ns: not significant.

**Table 2 genes-12-01778-t002:** Demographic and biochemical data of the 36 patients divided according to transferrin saturation. Data are expressed as median (1st–3rd percentiles), and number (percentage).

	Transferrin Saturation < 45%	Transferrin Saturation ≥ 45%	*p*
N°	25	11	
M/F	21/4	9/2	ns
Age (years)	54 (34–58)	45 (33.5–54)	ns
Hemoglobin (g/dL)	15.2 ^ (14.3–15.7)	14.5 ^^ (12.9–16)	ns
S-Iron (μg/dL)	97 (92–110)	199 (162.5–215.5)	<0.0001
Transferrin Saturation (%)	31.7 (27.1–36)	62.8 (56.4–75)	<0.0001
S-Ferritin (μg/L)	945 (745–1105)	1339 (844.5–2423.5)	0.057
LIC (μmol/g)	51.2 (39.0–69.2)	100.0 (52.9 –283.2)	<0.002
Body Mass Index (kg/m^2^)	22.6 (21.9–25.3)	23.5 (20.9–24.3)	ns
Glycemia (mg/dL)	87 (83.5–101)	90 (80.5–103)	ns
Triglycerides (mg/dL)	104 (77.5–122.8)	85 (63.3–106.8)	ns
HDL (mg/dL)	56 (49–64)	57 (46.5–60.5)	ns
ALT (U/L)	22 (16–27.5)	28 (26–41.3)	0.015

LIC: liver iron concentration was calculated based on MRI T2 * according to Galimberti et al. [15], and was available in 22 patients with transferrin saturation < 45% and all patients with transferrin saturation ≥ 45%. ^ Two patients with β-thalassemia trait; ^^ One patient with β-thalassemia trait; ns: not significant.

**Table 3 genes-12-01778-t003:** Age, hemoglobin, serum iron indices and liver iron concentration of the 10 patients carrying mutations in *SLC40A1*, *TFR2*, *HJV* and *HAMP*.

Patient	Sex	Age (years)	Hb (g/dL)	S-Iron (μg/dL)	Transferrin Saturation (%)	S-Ferritin (μg/L)	RMN T2* (msec)
1	M	25	15.2	116	31.7	1000	9
2	M	41	14.5	132	39.4	819	9.7
3	M	62	11.9	74	23.7	2600	14.29
4	F	21	12.1	108	38	1010	-†
5	M	59	13.6	143	36	1025	6.96
6	M	54	15.7	129	37	1908	6.5 ‡
7	M	12	15.8	297	90	151	19.5
8	M	46	16.1	183	83	3909	1.4
9	M	38	12.3	208	92	3510	1.02
10	F	19	14.4	272	87	1339	2.2

LIC: liver iron concentration was calculated based on MRI T2* according to Galimberti et al. [15]. † Iron depleted after the removal of 4 g of iron by phlebotomy therapy. ‡ Liver biopsy: Total Iron Score by Deugnier et al. [18]: 23 (Hepatic Iron Score 6,6,6; Sinusoidal Iron Score 1,2,2; Portal Iron Score 0,0,0).

**Table 4 genes-12-01778-t004:** In silico prediction of mutations’ effect on protein function.

Patient	Gene	cDNA— Protein Mutation	Protein Lenght	SIFT	Polyphen	Mutation Taster	Human Splicing Finder
1	*SLC40A1* (exon 5)	c.479T>C p.Val160Ala	not affected	affect protein function, score = 0.00	probably damaging score = 0.992	disease causing	-
2	*SLC40A1* (exon 6)	c.536G>C p. Arg179Thr s765023388	not affected	tolerated score = 0.57	probably damaging score = 0.996	disease causing; might affect splice site	alteration of an exonic ESE site; potential splicing alteration
3	*SLC40A1* (exon 7)	c.1049 C>A p.Ala350Asp	not affected	affect protein function, score = 0.00	probably damaging, score = 0.999	disease causing	-
4	*SLC40A1* (exon 8)	c.1481 G>A p.Gly494Asp	not affected	affect protein function, score = 0.00	probably damaging, score = 1.000	disease causing; might affect of splice site	no splicing site alteration
5	*SLC40A1* (5′ UTR)	c.-205A>C	not affected	-	-	disease causing	-
6	*SLC40A1* (exon 8)	c.1592T>C p. Val531Ala	not affected	affect protein function, score = 0.00	probably damaging, score = 1.000	disease causing; splice site change	alteration of an exonic ESE site; potential splicing alteration
7	*TFR2* (exon 5 and 13)	c.671T>G p.Leu224Arg	not affected	tolerated score = 0.53	possibly damaging, score = 0.8	protein features (might be) affected; splice site changes	alteration of an exonic ESE site; potential splicing alteration
		c.1540delG p.Asp514Metfs12	affected	affect protein function, score = 0.05	probably damaging, score = 0.987	disease causing; frameshift-splice site change	alteration of an exonic ESE site; potential splicing alteration
8	*TFR2* (exon 10 and 14)	c.1364G>A p.Arg455Gln [17] rs41303501	not affected	tolerated score = 0.29	possibly damaging, score = 0.837	protein features (might be) affected; mught affect splice site changes	activation of an exonic cryptic acceptor site and of one or more cryptic branch point(s); potential splicing alteration
		c.1640A>T p.Tyr547Phe rs573769443	not affected	affect protein function, score = 0.00	benign score = 0.238	protein features (might be) affected; might affect splice site changes	alteration of an exonic ESE site; potential splicing alteration
9	*HJV* (exon 3)	c.265 T>C p.Cys89Arg	not affected	affect protein function, score = 0.00	probably damaging score = 0.999	disease causing	no splicing alteration
		deletion: g.(?_144124721)_(144126704_?)del	-	-	-	-	-
10	*HAMP* (exon 3)	c.176 G>C p.Arg59Pro	not affected	tolerated score = 0.88	probably damaging score = 1.0	protein features (might be) affected; might affect splice site changes	activation of a cryptic acceptor site; potential splicing alteration

## Data Availability

Not applicale.

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
