# Peer review of "Identification of Novel Mutations by Targeted NGS Panel in Patients with Hyperferritinemia"

_genes, 2021, doi:10.3390/genes12111778_

Round 1

Reviewer 1 Report

The work presented by Giulia Ravasi et al., presents a Next Generation Sequencing targeted gene panel in patients with hyperferritinemia to identify genetic causes. Compared to already applied methods using single gene sequencing the advantage would be a quicker and cheaper large-scale sequencing approach. The authors could show with this approach that in 26 out of 36 patients’ genetic variants were not responsible for hyperferritinemia suggesting the existence of other genetic defects or gene-gene and gene-environment interactions needing further studies.

The work is well presented and there are only minor things that need the authors attention.

  1. The tables showing data could be rearranged to make it easier for reading. Maybe more spacing or another format could be helpful.
  2. The authors mention that other genetic defects, gene-gene, gene-environment interactions could also be responsible for hyperferritinemia, it would be nice if the authors could give some more explanations and examples in the discussion part to highlight the relevance for the reader.

Author Response

We thank the reviewer for her/his comments:

  1. We arrange the table accordingly, hoping they are now easier for the readers however it is also depend by editorial editing.
  2. We tried to better explain the point (see Conclusions section, page 11, lines 378-386). As far as other possible genetic defects still to be identified, we now mention, as an example, one recent study suggesting the existence of an inherited recessive trait explaining familial hyperferritinemia. By the other hand, we hypothesised that in some case hyperferritinemia and or iron overload are complex phenotypes. In complex diseases such as autism, hypertension and diabetes, analyzing a single locus is not enough to understand their pathophysiology and results in the so-called missing heritability problem. Identifying interaction effects between genes is one of the main tasks of genome‐wide association studies aiming to shed light on the biological mechanisms underlying complex diseases. Based on these assumptions we suggested that in some case hyperferritinemia and/or iron overload might be the result of complex gene-gene and gene-environment interactions, needing further studies.

Reviewer 2 Report

This manuscript by Giulia Ravasi and colleagues reports on novel mutations of the genes related to Haemochromatosis in patients with hyperferritinemia using NGS technology. The idea that identification of mutations using NGS is potentially interesting and important. However, the concept of this manuscript is a bit unclear for me. 

A few minor points:

(1) The authors analyzed the mutations in patients with non-HFE-related hyperferritinemia (lane 17). However, I missed the author’s points of selecting the patients, which is limited to the patients suspected of non-HFE HH, Ferroportin disease or inherited hyperfferritinemia (lane 89). I think this is very important for the concept of this study. I think that the brief description would be needed to explain why the authors set the exclusion criteria i (written in lane 92-94) for this study.

(2) Liver iron concentration was calculated in 22 patients with TSAT<45 that means exclusion of 3 patients? The description would be better addressed if there are any appropriate reasons.

(3) Analysis using gnomAD should be described in the Method section.

(4) Some genes were written in both full name and abbreviation (e.g. L-ferritin gene (FTL)), and others were only abbreviations (e.g. HFE, HAMP, TFR2 and SLC40A1). I think that full gene name and abbreviations should be spelled out on initial appearance of the manuscript or written in Abbreviations section.

Author Response

We thank the reviewer for her/his comments

  1. Hyperferritinemia with or without iron overload is a relatively common condition and many patients present to general medical practioners, hematologists, hepatologists and even to tertiary centres for disorders of iron metabolism for diagnosis. Many causes of hyperferritinemia are acquired and some might be genetically determined. As described in the introduction section, there are overlaps between different inherited and acquired forms of hyperferritinemia, requiring accurate evaluation to discriminate among the different causes of hyperferritinemia. Eventually, genetic test is offered to the patients according to the rules defined by dedicated Guidelines and reviews reported in the Reference section of the manuscript. Accordingly, subjects with forms of hyperferritinemia clearly attributable to acquired causes (inflammatory states, liver diseases, metabolic disorders ...) and patients with secondary iron overload diseases have to be excluded from genetic testing. We also excluded patients with mutations on HFE gene explaining the phenotype. This is because C282Y homozygosity and C282Y/H63D compound heterozygosity are the most common cause of hereditary hemochromatosis and testing for p.C282Y and p.H63D variants is rapid and less expensive than NGS testing that was then offered to selected patients. Hoping to have correctly understood the indications of the reviewer, we have now added a short period in the Introduction section (page 1, lines 40-42) and partially modified the Patients section (pages 2-3, lines 91-98).
  2. This is correct, three patients refused or were not able to perform MRI evaluation because of claustrophobia. We modified the text accordingly (page 3, line 112).
  3. We added information in the Method section (page 4, lines 147-150).
  4. Accordingly to Referee’s suggestion, we standardized genes’ spelling adding in the Abbreviations section.